# Registered report: Melanoma exosomes educate bone marrow progenitor cells toward a pro-metastatic phenotype through MET

**Jake Lesnik[1], Travis Antes[1], Jeewon Kim[2], Erin Griner[3], Luisa Pedro[4], Reproducibility Project: Cancer Biology***

[1]System Biosciences, LLC, Mountain View, United States; [2]Transgenic Research Center, Stanford University School of Medicine, Stanford, United States; [3]University of Virginia, Charlottesville, United States; [4]University of Cambridge, Cambridge, United Kingdom

***For correspondence:** tim@cos.io

**Group author details:**
Reproducibility Project: Cancer Biology See page 14

**Abstract** The Reproducibility Project: Cancer Biology seeks to address growing concerns about reproducibility in scientific research by conducting replications of selected experiments from a number of high-profile papers in the field of cancer biology. The papers, which were published between 2010 and 2012, were selected on the basis of citations and Altmetric scores (*Errington et al., 2014*). This Registered Report describes the proposed replication plan of key experiments from "Melanoma exosomes educate bone marrow progenitor cells toward a pro-metastatic phenotype through MET" by Peinado and colleagues, published in Nature Medicine in 2012 (*Peinado et al., 2012*). The key experiments being replicated are from Figures 4E, as well as Supplementary Figures 1C and 5A. In these experiments, Peinado and colleagues show tumor exosomes enhance metastasis to bones and lungs, which is diminished by reducing *Met* expression in exosomes (*Peinado et al., 2012*). The Reproducibility Project: Cancer Biology is a collaboration between the Center for Open Science and Science Exchange and the results of the replications will be published in *eLife*.

## Introduction

Exosomes are nanovesicles up to 100 nm in size that are derived from endosomal membranes and secreted by cells as a means of intercellular communication (*Mathivanan et al., 2010*). They contain a wide array of cargo including proteins, cytokines, and nucleic acids (*Kharaziha et al., 2012*). Recently, exosomes have been shown to play multiple roles in promoting carcinogenesis, including the regulation of metastatic niche formation, regulation of tumor immune response, and chemotherapeutic resistance (*Tickner et al., 2014*). Peinado and colleagues reported that exosomes derived from melanoma cells promoted metastasis through education of bone marrow-derived cells in order to prime the pre-metastatic niche and increase vascularization. They further showed that exosome-mediated metastasis was dependent on expression of MET in exosomes, and that MET protein was increased in exosomes found in patients with advanced melanoma (*Peinado et al., 2012*). MET is an oncogenic receptor tyrosine kinase that promotes proliferation, motility, and migration, and is often aberrantly activated in tumors (*Gherardi et al., 2012*; *Trusolino et al., 2010*). These findings indicate that exosomal MET may be a potential therapeutic target or biomarker for metastatic disease.

Supplementary Figure 1C characterizes exosomes isolated from B16-F10 melanoma cells using electron microscopy imaging and Western blotting for exosome protein markers. Supplementary

Figure 5A further characterizes these exosomes by assessing the levels of MET and pMET after shRNA-mediated depletion of *MET* in B16-F10 cells. These figures are essential to reproduce as they validate the expression of key proteins in exosomes that will subsequently be used to replicate Figure 4E. These experiments will be replicated in Protocols 1 and 2.

In Figure 4E, *Peinado et al. (2012)* reported that reduction of MET protein in melanoma exosomes reduced metastasis of B16-F10 melanoma cells to lung and bone (*Peinado et al., 2012*). Mice were pre-treated with exosomes isolated from B16-F10 melanoma cells expressing shRNAs directed against *Met* or control shRNAs, and then B16-F10 cells were injected subcutaneously. Primary tumor growth and metastasis to lungs and bone were assessed by luciferase imaging. This key experiment tests one of the central findings of the paper, namely that MET is necessary for melanoma exosome-mediated promotion of metastasis. This experiment will be replicated in Protocol 3.

## Materials and methods

### Protocol 1: Lentiviral knockdown of *Met* or non-silencing control in B16-F10 cells

This protocol utilizes shRNA to knock down *Met* in B16-F10 cells. This experiment generates key reagents (B16-F10 shMet and B16-F10 shScramble) that will subsequently be used in Protocols 2 and 3.

#### Sampling

- Experiment will be conducted once.
  - This experiment will generate B16-F10 shMet and B16-F10 shScramble stable cells.

Note: Information that these are stable transfectants was communicated by authors.

#### Materials and reagents

| Reagent | Type | Manufacturer | Catalog # | Comments |
|---|---|---|---|---|
| B16-F10 cells | Cell line | Original lab | n/a | From original lab |
| Dulbecco's modified Eagle's medium (DMEM) | Cell culture | VWR | 45000-30 | Communicated by authors |
| Fetal bovine serum (FBS) | Cell culture | GE Healthcare (HyClone) | SH30088.03 | Original catalog number not specified |
| Ultracentrifuge | Instrument | Sorval | | SureSpin 630 rotor |
| Thick wall ultracentrifuge tubes | Labware | Specific brand information will be left up to the discretion of the replicating lab and recorded later | | |
| 100X Penicillin/streptomycin | Cell culture | Life Technologies | 15140-155 | Communicated by authors |
| Phosphate buffered saline (PBS) | Cell culture | Fisher Scientific | MT-21-040-CM | Replaces VWR, cat# 45000-446; communicated by authors |
| Trypsin | Cell culture | Invitrogen | 25200-056 | Original brand not specified |
| pGIPZ mouse *Met* shRNA lentiviral particles (clone ID: V3LMM_456078) | Lentivirus | Dharmacon/GE Lifesciences | VGM5520-200377256 | Original from Thermo, which was acquired by Dharmcon/GE Lifesciences |
| pGIPZ non-silencing shRNA lentiviral particles | Lentivirus | Dharmacon/GE Lifesciences | RHS4348 | Original from Thermo, which was acquired by Dharmcon/GE Lifesciences |
| TransDux | Cell culture | System Biosciences | LV850A-1 | Replaces polybrene |
| Fluorescent microscope | Instrument | Leica | DMI300b | Original instrument not specified |
| Puromycin | Cell culture | Sigma-Aldrich | P8833-100MG | Included during communication with authors; original brand not specified |
| 150-mm tissue culture dish | Labware | E & K Scientific | EK-39160 | Original brand not specified |

## Procedure

Note:

- All cells will be sent for mycoplasma testing and short tandem repeat (STR) profiling.
- Cells maintained in DMEM supplemented with 10% exosome-depleted FBS, 100 U/ml penicillin and 100 µg/ml streptomycin at 37°C in a humidified atmosphere at 5% $CO_2$.

1. Deplete exosomes from FBS by ultracentrifugation at 100,000x$g$ for 70 min at 4°C.
2. Transduce B16-F10 cells with pGIPZ mouse *Met* shRNA (shMet) or pGIPZ non-silencing shRNA (sh Scramble) lentiviral particles following provider standard protocol (see Guide to Lentiviral Packaging and Transduction, System Biosciences) with a multiplicty of infection (MOI) of 10:1 (lentivirus particles:cells) and incubate overnight (16 hr).
   a. Combine culture medium with TransDux to a 1X final concentration.
3. The next day (16 hr later) replace media.
4. Determine transduction efficiency by green fluorescent protein (GFP) expression:
   a. Three d after transduction (when efficiency is anticipated to be near 80–90%), use fluorescent microscopy to image cells to determine transduction efficiency based on percent of GFP-positive cells.
      i. Use untransduced B16-F10 cells as GFP-negative cell population.
5. After determining transduction efficiency, replace media supplemented with 1.5 µg/ml of puromycin to select for transduced cells.
   a. Use untransduced B16-F10 cells as control for puromycin selection treatment.
6. Continue culture and passage of B16-F10 shMet and B16-F10 shScramble stable cells in puromycin for at least 28 d before further analysis.
   a. After initial selection of 28 d, B16-F10 shMet and B16-F10 shScramble cells should be maintained in puromycin, however when cells are plated for experiments they do not need puromycin added.
   b. Record detailed notes about culturing and passaging of cells, paying particular attention to density.

## Deliverables

- Data to be collected:
  - Detailed notes on cell culturing of both stable cell lines generated.
  - Transduction efficiency as a percentage of GFP-positive cells.
  - Fluorescent microscopy images of GFP-positive cells.
- Sample delivered for further analysis:
  - B16-F10 shMet and B16-F10 shScramble cells for exosome purification and Western blot analysis of METexpression for Protocol 2.
  - B16-F10 shMet and B16-F10 shScramble cells for exosome purification and mouse injection for Protocol 3.

## Confirmatory analysis plan

No analysis performed.

## Known differences from the original study

Similar to the original study, the transduction efficiency, determined by GFP expression, and the knockdown efficiency, determined by Western blot, will be measured (Protocol 2). The original protocol for selecting GFP-positive cells included a step to perform fluorescence-activated cell sorting (FACS) of the population to achieve a 95–99% GFP-positive population prior to puromycin selection (information provided by authors). The replication attempt will not include the FACS sorting step and instead will select the stable cells for at least 28 d before further analysis and will maintain the cell lines when not used in experimental procedures under puromycin selection. All known differences are listed in the Materials and reagents section above with the originally used item listed in the Comments section. All differences have the same capabilities as the original and are not expected to alter the experimental design.

## Provisions for quality control

The cell lines used in this experiment will undergo STR profiling to confirm their identity and will be sent for mycoplasma testing to ensure there is no contamination. Untransduced B16-F10 cells will be used to confirm the GFP-negative cell population and during puromycin selection to ensure efficient transduction occurs. Detailed cell culture notes will be recorded and made available to monitor growth rates of B16-F10 shScramble and B16-F10 shMet cells. All of the raw data will be uploaded to the project page on the Open Science Framework (OSF) (https://osf.io/ewqzf/) and made publically available.

## Protocol 2: Exosome purification and Western blot analysis of MET and phospho-MET expression

This protocol isolates exosomes from B16-F10 shScramble and B16-F10 shMet cells and then utilizes Western blot to characterize protein expression in cells generated in Protocol 1 and exosomes purified from this protocol. MET and pMET protein expression will be determined to verify MET knock down in exosomes, and exosome markers will also be assessed. This experiment will replicate Figures S5A and S1C (right panel).

## Sampling

- Experiment to be repeated a total of three times for a minimum power of 99%.
  - See Power calculations section for details.
- Each experiment has four cohorts:
  - Cohort 1: B16-F10 shScramble cells
  - Cohort 2: B16-F10 shMet cells
  - Cohort 3: Purified exosomes from B16-F10 shScramble cells
  - Cohort 4: Purified exosomes from B16-F10 shMet cells
- Each cohort is probed for:
  - HSC70 (exosome marker)
  - TSG101 (exosome marker)
  - CD63 (additional exosome marker)
  - MET
  - pMET Tyr1234/5
  - Glyceraldehyde-3-phosphate dehydrogenase (GAPDH; control)

## Materials and reagents

| Reagent | Type | Manufacturer | Catalog # | Comments |
|---------|------|-------------|-----------|----------|
| B16-F10 shScramble cells | Cell line | n/a | n/a | Generated in Protocol 1 |
| B16-F10 shMet cells | Cell line | n/a | n/a | Generated in Protocol 1 |
| Dulbecco's modified Eagle's medium (DMEM) | Cell culture | VWR | 45000-30 | Communicated by authors |
| Fetal bovine serum (FBS) | Cell culture | GE Healthcare (HyClone) | SH30088.03 | Original catalog number not specified |
| 100X Penicillin/streptomycin | Cell culture | Life Technologies | 15140-155 | Communicated by authors |
| Trypsin | Cell culture | Invitrogen | 25200-056 | Original brand not specified |
| 150 mm tissue culture dishes | Labware | E & K Scientific | EK-39160 | Original brand not specified |
| 10 mm tissue culture dishes | Labware | Fisher Scientific | 08-772-4F | Original brand not specified |
| 50 ml centrifuge tubes | Labware | Fisher Scientific | 14-959-49A | Original brand not specified |
| Ultracentrifuge | Instrument | Sorval | | SureSpin 630 rotor |
| Thick wall ultracentrifuge tubes | Labware | Specific brand information will be left up to the discretion of the replicating lab and recorded later | | |
| Phosphate buffered saline (PBS) | Cell culture | Fisher Scientific | MT-21-040-CM | Replaces VWR, cat# 45000-446; communicated by authors |

*Continued on next page*

*Continued*

| Reagent | Type | Manufacturer | Catalog # | Comments |
|---|---|---|---|---|
| Radioimmunoprecipitation assay (RIPA) buffer | Buffer | Sigma-Aldrich | R0278 | Original catalog number not specified |
| Complete protease inhibitor tablets | Inhibitor | Roche | 04693116001 | Original catalog number not specified |
| Bicinchoninic acid (BCA) protein determination kit | Reporter assay | Pierce | 23227 | Original catalog number not specified |
| 4X Laemmli sample buffer | Buffer | Bio-Rad | 161-0747 | Original brand not specified |
| Nanoparticle characterization system | Instrument | NanoSight | LM10 | |
| Nanoparticle Tracking Analysis | Software | NanoSight | Version 2.3 | Original version not specified |
| Sodium dodecyl sulfate polyacrylamide gel electrophoresis (SDS-PAGE) gradient gels | Western materials | BioRad | 456-1094 | Original catalog number not specified |
| Pre-stained protein molecular weight marker | Western materials | BioRad | 161-0377 | Original brand not specified |
| Electrophoresis buffer with SDS | Buffer | BioRad | 161-0772 | Original brand not specified |
| 10X Electrophoresis buffer Tris/glycine/SDS | Instrument | BioRad | 165-8004 | Original brand not specified |
| Methanol | Chemical | Fisher Scientific | A412-4 | Included during communication with authors; original brand not specified |
| Wet transfer system | Instrument | BioRad | 170-3930 | Original was a semi-dry system |
| Immobilon-P polyvinylidene difluoride (PVDF) membrane | Western materials | Millipore | IPVH10100 | Communicated by authors |
| SuperBlock | Chemical | Thermo Scientific | 37516 | Original was 2.5% non-fat milk |
| Tris-buffered saline (TBS) | Buffer | BioRad | 170-6435 | Included during communication with authors; original brand not specified |
| Tween-20 | Chemical | Fisher Scientific | BP337-500 | Included during communication with authors; original brand not specified |
| Mouse-anti-HSC70 | Antibodies | Stressgen | ALX-804-067 | Dilute 1:500; 70 kDa |
| Mouse anti-TSG101 | Antibodies | Santa Cruz | sc-7964 | Dilute 1:500; 45 kDa |
| Rabbit anti-CD63 | Antibodies | System Biosciences | EXOAB-CD63A-1 | Dilute 1:1,000; 53 kDa |
| Mouse anti-MET | Antibodies | Cell Signaling Technology | 3127 | Dilute 1:1,000; 145 kDa |
| Rabbit anti-pMET (Tyr1234/5) | Antibodies | Cell Signaling Technology | 3077 | Dilute 1:1,000; 145 kDa |
| Rabbit anti-GAPDH | Antibodies | Santa Cruz Biotechnology | sc-25778 | Dilute 1:1,000; 37 kDa |
| Sheep anti-mouse-HRP | Antibodies | GE Healthcare | NA931V | Dilute 1:4,000–1:20,000 |
| Donkey anti-rabbit-HRP | Antibodies | GE Healthcare | NA934V | Dilute 1:4,000–1:20,000 |
| Enhanced chemiluminescent (ECL) reagent | Western materials | Thermo Scientific | 34095 | Included during communication with authors; original brand not specified |
| ImageJ | Software | NIH | Version 1.48 | |

## Procedure

Note:

- All cells will be sent for mycoplasma testing and STR profiling.
- B16-F10 shMet and B16-F10 shScramble stable cells were generated in Protocol 1.
- All cells maintained in DMEM supplemented with 10% exosome-depleted FBS, 100 U/ml penicillin and 100 µg/ml streptomycin at 37°C in a humidified atmosphere at 5% $CO_2$.

- B16-F10 shMet and B16-F10 shScramble stable cells should have 1.5 µg/ml puromycin added to medium while maintaining the cell lines that are not used in the experimental procedure.

1. Deplete exosomes from FBS by ultracentrifugation at 100,000x*g* for 70 min at 4°C.
2. Grow B16-F10 shScramble and B16-F10 shMet cells for exosome purification and direct lysis.
   a. For exosome purification plate ~5 x 10$^6$ cells per 150 mm dish with 25 ml of media. Use two dishes per cell line.
   b. For direct lysis plate ~8–10 x 10$^6$ cells per 100 mm dish with 10 ml of media. Use one dish per cell line.
   c. B16-F10 shMet and B16-F10 shScramble cells do not need puromycin added to plates used in the experimental procedure.
3. Grow exponentially until cells reach 80–90% confluence.
   a. Culture 100 mm dishes overnight.
   b. Culture 150 mm dishes for 48–72 hr.
4. Directly lyse in 100 mm dishes:
   a. Wash cells in PBS.
   b. Prepare RIPA buffer and harvest cells directly in lysis buffer.
      i. Add a complete protease inhibitor to RIPA buffer.
   c. Clear lysates by benchtop centrifugation at 14,000x*g* for 20 min at 4°C.
   d. Measure protein concentration of supernatant with a BCA kit following manufacturer's instructions.
   e. Prepare 30 µg of total protein per sample with 4X Laemmli buffer.
      i. Store at -20°C until analysis.
5. Purify exosomes from 150 mm dishes:
   a. Collect supernatant from cell cultures in 50 ml centrifuge tubes.
   b. Pellet cells from supernatant in a benchtop centrifuge at 500x*g* for 10 min at 4°C.
   c. Transfer supernatant to thick wall ultracentrifuge tubes.
   d. Remove cell debris by ultracentrifugation at 20,000x*g* for 20 min at 4°C.
   e. Collect supernatant.
   f. Harvest exosomes by ultracentrifugation at 100,000x*g* for 70 min at 4°C.
   g. Resuspend the exosome pellet in 20 ml of PBS.
   h. Pellet exosomes by ultracentrifugation at 100,000x*g* for 70 min at 4°C.
   i. Resuspend exosome pellet in 100 µl PBS.
      i. Exosomes can be stored at -20°C for 2–3 weeks.
   j. Measure protein concentration with a BCA kit following manufacturer's instructions.
      i. Estimated yield of exosomes should be around 1–2 µg/1 x 10$^6$ cells.
   k. Characterize exosome pellet using standard Nanosight NTA analysis.
      i. 10–20 µg of exosome protein is needed for analysis.
      ii. Report on size distribution and concentration of exosomes.
   l. Prepare 30 µg of total protein per sample with 4X Laemmli buffer.
      i. Store at -20°C until analysis.
6. Load 30 µg of total protein with sample buffer in each lane on an SDS-PAGE gel with a protein molecular weight marker.
7. Run electrophoresis at constant voltage (100 V) for ~1–2 hr in 1X electrophoresis buffer following manufacturer instructions.
8. Transfer gel to membrane using a wet transfer system at constant amperage (350 mA) for 45 min in 1X Tris-glycine buffer supplemented with 20% methanol.
   a. Pre-soak membrane with methanol and then 1X transfer buffer before assembly.
9. Block membranes in SuperBlock following manufacturer's instructions.
10. Probe membranes with the following primary antibodies overnight at 4°C diluted in SuperBlock.
    a. mouse anti-HSC70; use at 1:500; 70 kDa
    b. mouse anti-TSG101; use at 1:500; 45 kDa
    c. anti-CD63; use at 1:1000; 53 kDa
    d. mouse anti-MET; use at 1:1000; 145 kDa
    e. rabbit anti-pMET Tyr1234/5; use at 1:1000; 145 kDa
    f. rabbit anti-GAPDH; use at 1:1000; 37 kDa
11. Wash membranes three times 10 min in Ticket Tax Box Service (TTBS).
    a. TTBS = 1X TBS supplemented with 0.1% tween.

12. Detect primary antibodies with the following secondary antibodies diluted in SuperBlock for 1 hr.
     a. sheep anti-mouse-HRP; use at 1:4000–1:20,000
     b. donkey anti-rabbit-HRP; use at 1:4000–1:20,000
13. Wash membranes three times for 10 min each in TTBS.
14. Detect using ECL reagent following manufacturer's instructions.
15. Quantify intensities of the immunoreactive bands by densitometry.
     a. Normalize total MET to GAPDH.
     b. Normalize pMET (Tyr1234/1235) to GAPDH.
16. Repeat independently two additional times.

## Deliverables

- Data to be collected:
  - Exosome characterization data (including protein concentration using a BCA kit).
  - Images of probed membranes (full images with ladder) (compare to Figure S1C [right panel]).
  - Quantified levels of total MET and Phospho-MET normalized to GAPDH for all conditions (compare to Figure S5A).

## Confirmatory analysis plan

This experiment assesses if knockdown of *Met* alters total MET and phospho-MET expression in cells and exosomes. This replication attempt will perform the following statistical analysis:

- Statistical analysis:
  - One-way multivariate analysis of variance (MANOVA) comparing the relative mean of photon signal for total MET and phospho-MET normalized to GAPDH from B16-F10 shScramble and B16-F10 shMet cells.
    - Planned comparisons with the Bonferroni correction:
      1. Total MET in B16-F10 shScramble cells compared with B16-F10 shMet cells
      2. Phospho-MET in B16-F10 shScramble cells compared with B16-F10 shMet cells
  - One-way MANOVA comparing the relative mean of photon signal for total MET and phospho-MET normalized to GAPDH from B16-F10 shScramble and B16-F10 shMet exosomes.
    - Planned comparisons with the Bonferroni correction:
      1. Total MET in B16-F10 shScramble exosomes compared to B16-F10 shMet exosomes
      2. Phospho-MET in B16-F10 shScramble exosomes compared to B16-F10 shMet exosomes
- Meta-analysis of effect sizes:
  - Compute the effect sizes of each comparison, compare them against the reported effect size in the original paper, and use a meta-analytic approach to combine the original and replication effects, which will be presented as a forest plot.

## Known differences from the original study

The cell lines when not used in experimental procedures will be maintained under puromycin selection. The replication attempt will include an additional exosome marker, CD63, not included in the original study. All known differences are listed in the Materials and reagents section above with the originally used item listed in the Comments section. All differences have the same capabilities as the original and are not expected to alter the experimental design.

## Provisions for quality control

The cell lines used in this experiment will undergo STR profiling to confirm their identity and will be sent for mycoplasma testing to ensure there is no contamination. This protocol analyzes the knockdown efficiency of c-Met in B16-F10 cells. Isolated exosomes are characterized by NanoSight and

are analyzed for typical exosome markers by Western blot, including CD63, an additional marker not included in the original study. All of the raw data, including the image files and quantified bands from the Western blot, will be uploaded to the project page on the OSF (https://osf.io/ewqzf/) and made publically available.

## Protocol 3: Exosome-dependent MET signaling on primary tumor growth and metastasis

This experiment tests the effect of exosome-derived MET on primary growth and metastasis of melanoma cells. This is a replication of Figure 4E, which assesses metastasis in lungs and bone using bioluminescent imaging.

### Sampling

- ▪ Experiment will use seven mice per cohort for a minimum power of 82%.
  - • See Appendix for detailed power calculations
- ▪ Each experiment has three cohorts:
  - • Cohort 1: Synthetic unilamellar liposomes injected into C57BL/6 mice
  - • Cohort 2: B16-F10 shScramble exosomes injected into C57BL/6 mice
  - • Cohort 3: B16-F10 shMet exosomes injected into C57BL/6 mice

### Materials and reagents

| Reagent | Type | Manufacturer | Catalog # | Comments |
|---|---|---|---|---|
| B16-F10 shScramble cells | Cell line | n/a | n/a | Generated in Protocol 1 |
| B16-F10 shMet cells | Cell line | n/a | n/a | Generated in Protocol 1 |
| Dulbecco's modified Eagle's medium (DMEM) | Cell culture | VWR | 45000-30 | Communicated by authors; for B16-F10 shScramble and shMet cells |
| DMEM | Cell culture | Life Technologies | 11995-040 | Communicated by authors; for B16-F10-luciferase cells |
| Fetal bovine serum (FBS) | Cell culture | GE Healthcare (HyClone) | SH30088.03 | Original catalog number not specified |
| 100X Penicillin/streptomycin | Cell culture | Life Technologies | 15240-062 | Communicated by authors |
| 150 mm tissue culture dish | Labware | E & K Scientific | EK-39160 | Original brand not specified |
| 10 mm tissue culture dish | Labware | Fisher Scientific | 08-772-4F | Original brand not specified |
| T75 tissue culture flasks | Labware | Sigma-Aldrich | CLS430641 | Originally not specified |
| T25 tissue culture flasks | Labware | Sigma-Aldrich | C6356 | Originally not specified |
| 50 ml centrifuge tubes | Labware | Fisher Scientific | 14-959-49A | Original brand not specified |
| Ultracentrifuge | Instrument | Sorval | | SureSpin 630 rotor |
| Thick wall ultracentrifuge tubes | Labware | Specific brand information will be left up to the discretion of the replicating lab and recorded later | | |
| Phosphate buffered saline (PBS) | Buffer | Fisher Scientific or Life Technologies | MT-21-040-CM or 70011-044 | Replaces VWR, cat# 45000-446 Communicated by authors |
| Trypsin | Cell culture | Invitrogen or Life Technologies | 25200-056 or 12604-021 | Original brand not specified |
| Bicinchoninic acid (BCA) protein determination kit | Reporter assay | Pierce | 23227 | Original catalog number not specified |
| Nanoparticle characterization system | Instrument | NanoSight | LM10 | |
| Nanoparticle Tracking Analysis | Software | NanoSight | Version 2.3 | Original version not specified |

*Continued on next page*

*Continued*

| Reagent | Type | Manufacturer | Catalog # | Comments |
|---------|------|--------------|-----------|----------|
| Synthetic unilamellar 100 nm liposomes | Chemical | Encapsula NanoSciences | n/a | Composition: 13 mg/ml L-α-phosphatidylcholine, 2.78 mg/ml cholesterol (7:3 molar ratio P:C); communicated by authors |
| 6-week-old C57BL/6 female mice | Animal model | Charles River | Strain code: 027 | Replaces Jackson Laboratories used in the original study; age communicated by authors |
| B16-F10-luciferase cells | Cell line | Original lab | n/a | From original lab |
| 1 ml syringe with Luer-Lok tip | Labware | Fisher Scientific | 14-823-30 | Original brand not specified |
| Needle | Labware | Specific brand information will be left up to the discretion of the replicating lab and recorded later | | |
| 3/10 cc syringe with 29G1/2 attached needle | Labware | Terumo Medical Products | SS30M2913 | Original brand not specified |
| AErrane (Isoflurane) | Chemical | Baxter | n/a | Replaces original from Baxter, catalog # 400-326-09; communicated by authors |
| Oxygen | Chemical | Praxair | TC 3AAM 154 | Replaces original from Airgas; communicated by authors |
| D-luciferin, potassium salt | Reporter assay | Biosynth | L-8220 | Replaces original brand from Life Technologies; communicated by authors |
| IVIS Spectrum system | Instrument | Xenogen (Caliper) | | |
| Living Image software | Software | Xenogen (Caliper) | Version 4.2 | |

Note:

- All cells will be sent for mycoplasma testing and STR profiling.
- B16-F10 shMet and B16-F10 shScramble stable cells were generated in Protocol 1.
- All cells maintained in DMEM supplemented with 10% exosome-depleted FBS, 100 U/ml penicillin and 100 µg/ml streptomycin at 37°C in a humidified atmosphere at 5% $CO_2$.
- B16-F10 shMet and B16-F10 shScramble stable cells should have 1.5 µg/ml puromycin added to medium while maintaining the cell lines that are not used in the experimental procedure.
- The original study indicated mice were 8–10 weeks old, however the authors clarified that the mice were 6 weeks old.

1. Deplete exosomes from FBS by ultracentrifugation at 100,000x$g$ for 70 min at 4°C.
2. Two–three days before needing exosomes, plate ~5 x $10^6$ B16-F10 shScramble and B16-F10 shMet cells per 150 mm dish with 25 ml of media for exosome purification.
   a. Use three to six dishes per cell line to obtain needed amount of exosomes.
   b. B16-F10 shMet and B16-F10 shScramble cells do not need puromycin added to plates used in the experimental procedure.
3. Culture cells exponentially for 48–72 hr until cells reach 80–90% confluence.
4. Purify exosomes from 150 mm dishes:
   a. Collect supernatant from cell cultures in 50 ml centrifuge tubes.
   b. Pellet cells from supernatant in a benchtop centrifuge at 500x$g$ for 10 min at 4°C.
   c. Transfer supernatant to thick wall ultracentrifuge tubes.
   d. Remove cell debris by ultracentrifugation at 20,000x$g$ for 20 min at 4°C.
   e. Collect supernatant.
   f. Harvest exosomes by ultracentrifugation at 100,000x$g$ for 70 min at 4°C.
   g. Resuspend the exosome pellet in 20 ml of PBS.
   h. Pellet exosomes by ultracentrifugation at 100,000x$g$ for 70 min at 4°C.
   i. Resuspend exosome pellet in 100 µl PBS.
      i. An aliquot of the exosome preparation can be stored at -20°C until Nanosight analysis.
   j. Measure protein concentration with a BCA kit following manufacturer's instructions.

 i. Estimated yield of exosomes should be around 1–2 µg/1x10$^6$ cells.

 k. Characterize exosome pellets using standard Nanosight NTA analysis.

 i. An aliquot of each exosome preparation will be stored at -20°C and analyzed all at once following the final preparation.

 ii. 10–20 µg of exosome protein is needed for analysis.

 iii. Report on size distribution and concentration of exosomes.

 iv. Report the number of exosomes per µg protein (as measured by BCA) for each sample.

 l. Prepare samples at a concentration of 50 ng/µl to achieve 5 µg of total protein diluted in 100 µl of PBS.

5. Following protein quantification of each preparation, inject intravenously, via retro-orbital injection, freshly isolated B16-F10 shScramble or B16-F10 shMet exosomes, or synthetic uni-lamellar 100 nm liposomes into 6-week-old C57BL/6 female mice three times a week for a total of 28 d.

 a. Sample injection schedule:

 i. Each cohort will be injected on Monday, Wednesday, and Friday with a fresh exo-some preparation (>35 µg total) each week for a total of 4 weeks.

 ii. Step 4 will be performed for each injection day.

 b. It is crucial to inject fresh exosomes every time, do not freeze down, and inject right after purification and quantification following steps 2–4 above.

 c. Volume of injection is 100 µl.

 d. Amount of synthetic liposomes injected, 1.25 µg of L-$\alpha$-phosphatidylcholine (PC) will mimic 5 µg of exosome protein, which is based on a theoretical 4:1 protein:PC ratio (communicated by authors).

 i. Dilute 1.92 µl of a 1:20 dilution of the stock concentration of synthetic liposomes into 100 µl PBS for each injection.

6. After 28 d, inject $1 \times 10^6$ B16-F10-luciferase cells diluted in 100 µl of PBS subcutaneously in the flank of mice.

7. Measure primary tumor volume three times a week for a total of 21 d.

 a. Use calipers to measure width and height with volume determined as (length x width$^2$)/2. (additional recorded information)

 i. Note: tumor volume detection will be limited to <1000 mm$^3$.

 b. Record latency. (additional recorded information)

 i. Note: Perform Steps 8 through 10 (luciferin injection, euthanasia, dissection, and imaging) from mice from different cohorts in parallel (i.e., one from each of the three cohorts) so variation during the procedure is equal across cohorts. (additional detail)

8. After 21 d anesthetize mice and inject 50 mg/kg of D-luciferin via retro-orbital injection in 100–200 µl PBS (volume depending on the body weight).

 a. Weigh mice.

 b. Use isoflurane and O$_2$ to anesthetize mice.

9. Five min later euthanize mice by cervical dislocation under anesthesia and dissect tissues (lungs and femurs).

 a. Dissect out primary tumors and record weight (additional parameter).

10. Image dissected primary lungs and bones (femurs) for luciferase expression in IVIS Imaging system

 a. Record the time from euthanasia to imaging for each mouse.

 b. Record photon flux.

 a. Take two–three exposure times for each sample.

 i. Use same exposure time for each tissue from all mice during analysis.

## Deliverables

- Data to be collected:
  - Mouse health records (injection schedule, time from tumor cell injection to detectable tumors [latency], weight of mice at end of experiment, mortality report)
  - Exosome characterization data (including protein concentration using a BCA kit).
  - Raw numbers and calculated tumor volume for all mice, and graph of tumor volume ver-sus time for all conditions during course of treatment.

- Time of euthanasia to imaging for each mouse.
- Images of lungs and bones for luciferase expression (compare to Figure 4E).
- Raw photon flux values of all analyzed images of each tissue for all conditions using the same exposure time (compare to Figure 4E).
- Final weight of tumors.

## Confirmatory analysis plan

This experiment assesses if knockdown of Met alters primary tumor growth and metastasis. This replication attempt will perform the following statistical analysis:

- Statistical Analysis:
  - Bonferroni corrected one-way ANOVAs comparing lung photon flux of mice treated with synthetic unilamellar liposomes, or exosomes from B16-F10 shScramble or B16-F10 shMet cells and the following planned comparisons with the Fisher's least significant difference (LSD) test:
    1. Synthetic unilamellar liposomes compared to B16-F10 shScramble exosomes
    2. B16-F10 shMet exosomes compared to B16-F10 shScramble exosomes
  - Bonferroni corrected Kruskal–Wallis test comparing bone photon flux of mice treated with synthetic unilamellar liposomes, or exosomes from B16-F10 shScramble or B16-F10 shMet cells and the following planned comparisons (Wilcoxon–Mann–Whitney test) with the Fisher's LSD test:
    1. Synthetic unilamellar liposomes compared to B16-F10 shScramble exosomes
    2. B16-F10 shMet exosomes compared to B16-F10 shScramble exosomes
- Meta-analysis of effect sizes:
  - Compute the effect sizes of each comparison, compare them against the reported effect size in the original paper and use a meta-analytic approach to combine the original and replication effects, which will be presented as a forest plot.
- Exploratory analysis:
  - Comparison of primary tumor growth rates.
    - This is exploratory analysis. We will measure tumor growth rates across all mouse cohorts over the length of the study. These data were collected, but not reported in the original study, and found to not be different. We will plot growth curves and calculate area under the curve for each mouse. We will then perform a one-way analysis of variance (ANOVA) analysis, with the following planned comparisons with the Fisher's LSD test:
      1. Synthetic unilamellar liposomes compared to B16-F10 shScramble exosomes
      2. B16-F10 shMet exosomes compared to B16-F10 shScramble exosomes
  - Comparison of final primary tumor weights.
    - This is exploratory analysis. We will measure tumor weights across all mouse cohorts at the end of the study. These data were not reported in the original study. We will perform a one-way ANOVA analysis, with the following planned comparisons with the Fisher's LSD test:
      1. Synthetic unilamellar liposomes compared to B16-F10 shScramble exosomes
      2. B16-F10 shMet exosomes compared to B16-F10 shScramble exosomes

## Known differences from the original study

The cell lines when not used in experimental procedures will be maintained under puromycin selection. The number of exosomes injected (based on protein content) will be reported for each preparation from each cohort. The original data on primary tumor growth were not shown and final primary tumor weights were not recorded. The replication attempt will record and present these data as well as tumor latency. All known differences are listed in the Materials and reagents section above with the originally used item listed in the Comments section. All differences have the same capabilities as the original and are not expected to alter the experimental design.

## Provisions for quality control

The cell lines used in this experiment will undergo STR profiling to confirm their identity and will be sent for mycoplasma testing to ensure there is no contamination. Isolated exosomes will be injected

immediately after protein quantification and will be characterized by NanoSight following the final preparation to ensure the integrity of the samples. Exosomes and luciferin will be injected intravenously, via retro-orbital injection, similar to the original study. While it will be attempted to be the same for all animals, the time from euthanasia to imaging for each mouse will be recorded as an additional quality control parameter. All of the raw data, including the image files and photo flux values, will be uploaded to the project page on the OSF (https://osf.io/ewqzf/) and made publically available.

## Power calculations

For additional details on power calculations, please see analysis scripts and associated files on the OSF:

https://osf.io/nyb8d/

## Protocol 1
Not applicable

## Protocol 2
Summary of original data reported in Figure S5A (calculated from data shared by original authors):

| Dataset being analyzed | | Mean | SD | N |
|---|---|---|---|---|
| Total MET | B16-F10 shScramble cells | 1.000 | 0.0085 | 2 |
| | B16-F10 shMet cells | 0.641 | 0.0254 | 2 |
| Phospho-MET | B16-F10 shScramble cells | 1.000 | 0.0318 | 2 |
| | B16-F10 shMet cells | 0.234 | 0.0134 | 2 |

The replication will also include analysis of total MET and phospho-MET in exosomes from B16-F10 shScramble and B16-F10 shMet cells. We will use the same calculated sample size as determined for the cells with the assumption that the cells and exosomes have similar values, which Peinado and colleagues (2012) present in Figure 4A.

Test family

- two-tailed *t* test, difference between two independent means, Bonferroni's correction: alpha error = 0.025

Power calculations performed with G*Power software, version 3.1.7 (*Faul et al., 2007*).
Total Met:

| Group 1 | Group 2 | Effect size *d* | A priori power | Group 1 sample size | Group 2 sample size |
|---|---|---|---|---|---|
| B16-F10 shScramble | B16-F10 shMet | 18.97367 | 99.9% | 2* | 2* |

* A minimum sample size of three per group will be used making the power 99.9%

Phospho-Met:

| Group 1 | Group 2 | Effect size *d* | A priori power | Group 1 sample size | Group 2 sample size |
|---|---|---|---|---|---|
| B16-F10 shScramble | B16-F10 shMet | 31.38411 | 99.9% | 2[†] | 2[†] |

[†] A minimum sample size of three per group will be used making the power 99.9%.

## Protocol 3

Summary of original data reported in Figure 4E (calculated from data shared by original authors):

| Dataset being analyzed | | Mean | SD | N |
|---|---|---|---|---|
| Lung photon flux | Synthetic unilamellar liposomes | 23,386 | 9,138 | 7 |
| | Exosomes from B16-F10 shScramble cells | 50,771 | 14,966 | 7 |
| | Exosomes from B16-F10 shMet cells | 12,667 | 9,032 | 7 |
| Bones photon flux | Synthetic unilamellar liposomes | 0 | 0 | 5 |
| | Exosomes from B16-F10 shScramble cells | 44,660 | 32,595 | 5 |
| | Exosomes from B16-F10 shMet cells | 0 | 0 | 5 |

Lung photo flux:
Test family

- ANOVA: Fixed effects, omnibus, one-way, Bonferroni's correction: alpha error = 0.025.

Power calculations performed with G*Power software, version 3.1.7 (*Faul et al., 2007*). ANOVA F-test statistic performed with R software, version 3.1.2 (*Team, 2014*). Partial $\eta^2$ calculated from (*Lakens, 2013*).

| Dataset | Groups | F-test statistic | Partial $\eta^2$ | Effect size $f$ | A priori power | Total sample size |
|---|---|---|---|---|---|---|
| Lung | Synthetic unilamellar liposomes, B16-F10 shScramble, B16-F10 shMet | $F(2,18) = 20.8417$ | 0.69841 | 1.52176 | 94.8%* | 12 total mice* (3 groups) |

* 21 total mice will be used based on the bones photon flux planned comparison calculations making the power 99.9%.

Test family

- 2 tailed *t* test, difference between two independent means, Fisher's LSD correction: alpha error = 0.025.

Power calculations performed with G*Power software, version 3.1.7 (*Faul et al., 2007*).

| Group 1 | Group 2 | Effect size $d$ | A priori power | Group 1 sample size | Group 2 sample size |
|---|---|---|---|---|---|
| Synthetic unilamellar liposomes | B16-F10 shScramble | 2.20861 | 86.1% | 6* | 6* |
| B16-F10 shScramble | B16-F10 shMet | 3.08277 | 87.6% | 4[†] | 4[†] |

* 7 mice will be used per group based on the bones photon flux calculations making the power 92.5%.
[†] 7 mice will be used per group based on the bones photon flux calculations making the power 99.8%.

Bones photon flux:
Test family

- ANOVA: Fixed effects, omnibus, one-way, Bonferroni's correction: alpha error = 0.025.

Power calculations performed with G*Power software, version 3.1.7 (*Faul et al., 2007*). ANOVA F-test statistic performed with R software, version 3.1.2 (*Team, 2014*). Partial $\eta^2$ calculated from (*Lakens, 2013*).

| Dataset | Groups | F-test statistic | Partial $\eta^2$ | Effect size $f$ | A priori power | Total sample size |
|---|---|---|---|---|---|---|
| Bones | Synthetic unilamellar liposomes, B16-F10 shScramble, B16-F10 shMet | $F(2,12) = 9.3866$ | 0.61005 | 1.25077 | 82.2%* | 12* (3 groups) |

\* Since the nonparametric Kruskal–Wallis test will be performed for the analysis instead of an ANOVA, a 15% adjustment in sample size (15 instead of 12) is taken into account. A total of 21 mice will be used based on the bones photon flux planned comparison calculations making the power 98.1% (using a 15% adjusted sample size of 18 instead of 21).

Test family

- 2 tailed $t$ test, Wilcoxon–Mann–Whitney test (two groups), Fisher's LSD correction: alpha error = 0.025.

Power calculations performed with G*Power software, version 3.1.7 (*Faul et al., 2007*).

| Group 1 | Group 2 | Effect size $d$ | A priori power | Group 1 sample size | Group 2 sample size |
|---|---|---|---|---|---|
| Synthetic unilamellar liposomes | B16-F10 shScramble | 1.93769 | 81.6% | 7 | 7 |
| B16-F10 shScramble | B16-F10 shMet | 1.93769 | 81.6% | 7 | 7 |

## Acknowledgements

The Reproducibility Project: Cancer Biology core team would like to thank the original authors, in particular Hector Peinado, for generously sharing critical information as well as reagents to ensure the fidelity and quality of this replication attempt. We thank Courtney Soderberg at the Center for Open Science for assistance with statistical analyses. We thank Maureen Peterson at System Biosciences for help with some protocol details. We would also like to thank the following companies for generously donating reagents to the Reproducibility Project: Cancer Biology: American Type Culture Collection (ATCC), Applied Biological Materials, BioLegend, Charles River Laboratories, Corning Incorporated, DDC Medical, EMD Millipore, Harlan Laboratories, LI-COR Biosciences, Mirus Bio, Novus Biologicals, Sigma-Aldrich, and System Biosciences (SBI).

## Additional information

### Group author details

Reproducibility Project: Cancer Biology

Elizabeth Iorns: Science Exchange, Palo Alto, United State; William Gunn: Mendeley, London, United Kingdom; Fraser Tan: Science Exchange, Palo Alto, United States; Joelle Lomax: Science Exchange, Palo Alto, United States; Nicole Perfito: Science Exchange, Palo Alto, United States; Timothy Errington: Center for Open Science, Charlottesville, United States

### Competing interests

JL: System Biosciences Inc., is a Science Exchange associated lab. System Biosciences produces some of the reagents used in this study, specifically TransDux, and the rabbit anti-CD63 antibody. TA: System Biosciences Inc., is a Science Exchange associated lab. System Biosciences produces some of the reagents used in this study, specifically TransDux, and the rabbit anti-CD63 antibody. JK: This is a Science Exchange associated lab. The other authors declare that no competing interests exist.

## Funding

| Funder | Author |
| --- | --- |
| Laura and John Arnold Foundation | Reproducibility Project: Cancer Biology |

The funders had no role in study design, data collection and interpretation, or the decision to submit the work for publication.

## Author contributions

JL, TA, EG, LP, Drafting or revising the article; JK, Transgenic Research Center is a Science Exchange associated lab., Drafting or revising the article; RP:CB, Conception and design, Drafting or revising the article

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
