## [Decision Letter]

Thank you for submitting your work entitled "Registered report: Melanoma exosomes educate bone marrow progenitor cells toward a pro-metastatic phenotype through MET" for peer review at *eLife*. Your submission has been favorably evaluated by Charles Sawyers (Senior Editor), Michael Green (Reviewing Editor), and four reviewers, one of whom, John Harris, has agreed to reveal his identity.

The reviewers have discussed the reviews with one another and the Reviewing Editor has drafted this decision to help you prepare a revised submission.

Although the reviewers were generally positive there were several substantive concerns that need to be addressed in a revised registered report. Of particular concern is the method used for exosome isolation (point 1) and the apparent conflict of interest (point 2).

Essential revisions:

1) Given the aim of this project, it is of paramount importance that all details of the experimental design be reproduced as described in Peinado et al., Nature Medicine, 2012. In particular, there are some critical differences in the experimental procedures of the proposed study that may lead to big differences in the results compared to the original Peinado et al. study. First and foremost, the method proposed to isolate exosomes for western blot and functional studies, as well as for the depletion of exosomes from FBS is based on ExoQuick extraction rather than sequential ultracentrifugation as in the original publication. This is a major issue that can hinder the reproducibility of the data, as during exosome isolation using ExoQuick, contaminants (e.g. VEGF) are trapped by ExoQuick, therefore education would not be strictly and solely due to exosomes. Clearly the authors are aware of this issue, as they have highlighted this difference. The main problem is that 1 μg of exosomal protein from ExoQuick-isolated exosomes would be the equivalent of 0.1-0.5 μg of exosomes isolated by classical ultracentrifugation (UC), as in the Peinado et al. study. This 10-fold difference in the amount of exosome protein delivered in education studies would lead to "under-education". Since in the Peinado et al. study the exosomal preparations were free of contaminants, as demonstrated by electron microscopy and western blot, and the authors' likely carry significant contaminants, a side by side comparison of ExoQuick and ultracentrifugation would be required to determine the amount of ExoQuick exosomal preparation equivalent to 10 μg of exosomes. A reference is not sufficient, actual data needs to be presented for this comparison, including quantitation and electron microscopy studies. The authors would have to measure the yield of exosomes, proteins and contaminants from both methods in parallel.

There are numerous publications presenting data comparing exosomal isolation methods that support the use of ultracentrifugation versus polymer-based precipitating methods such as ExoQuick. In particular, three papers demonstrate that up to 23 times more material is precipitated with ExoQuick compared to standard serial UC (Van Deun J et al., Journal of Extracellular Vesicles, 2014, 18(3), PMID: 25317274; Zlotogorski-Hurvitz A et al., J Histochem Cytochem. 2015, 63(3): 181-9, PMID: 25473095; Yamada T et al., J Vet Med Sci. 2012, 74(11):1523-5, PMID: 22785357). A recent study provides electron microscopy pictures that demonstrate that the higher protein yield obtained with the ExoQuick precipitation method compared to the UC gold standard is due to contaminants such as large protein aggregates (Sáenz-Cuesta M et al., Front Immunol, 2015, 6:50, PMID: 25762995).

2) Importantly, the fact that the first three authors on the proposed Reproducibility project study are employees of System Biosciences, the company that produces ExoQuick as well as ExoFBS raises a red flag and creates a huge conflict on interest in terms of the choice of methodology. The choice of experimental approaches and reagents should be driven by the desire to perform the tests in conditions as close as possible to the original study, but the authors are clearly choosing a critical reagent that introduces a large variation from the original study based on financial interests.

3) It is not clear whether the authors plan to use fresh or frozen (-20°C) exosomal preparations for education. There have been studies that have shown that freezing exosomes results in lysis of more than 50% of the preparation (Oosthuyzen et al., J Physiol. 2013, 591(Pt 23): 5833-5842) which will affect the amount delivered in a single dose (if the exosomes were quantified prior to freezing and that amount is used for calculation of material to inject for education). Freezing could also affect the functionality of exosomes. The original Peinado et al. study had performed the education studies with fresh exosomes.

4) Another major problem is that the authors are using non-transduced B16-F10 cells as control for the B16-F10 shMET. It is highly recommended that a scrambled short hairpin control (shScramble) be used, as it is widely known that infection and puromycin treatment of cells may alter the cell population and the appropriate scrambled control in the same vector is available from the same company that produces the shMET. Importantly, the authors plan to rely solely on puromycin resistance to select the knockdown clones, but this is not sufficient to isolate a pure population of transduced cells that maintains high levels of vector integration and expression. In addition to verifying knockdown of MET, the authors also need to show that Met knockdown did not affect the growth of the cells (every integration event is unique and you never know where the lentivirus integrated).

5) In protocol 1: The authors propose to use puromycin selection to generate stable B16-F10 shMet transfectants instead of FACS sorting GFP-positive cells, which should be sufficient. Upon reviewing the original paper, we could not find any details how the authors generated B16-F10 shMet cells, and whether these were stable transfectants. Was this communicated directly by the authors of the study?

6) It looks like they never tested whether MET was downregulated in exosomes themselves in supplementary Figure 5a, just in the parent cells from which they purified the exosomes. This is planned for the B16-F10 control and shMET exosomes, which we think is good. We would also suggest including B16-F1 exosomes, as there was a comparison by Western blot for the B16-F10 to B16-F1 exosomes in Figure 4a, at least that would be something to compare across studies.

7) In the original paper there is a microarray analysis where genes with a fold change greater than 2 are reported. Apparently p-values from a t-test were computed with a permutation approach but not reported. It would be interesting to repeat also this experiment reporting the statistical significant genes controlling the false discovery rate.

8) In the statistical analyses plan of the report it is claimed that Fisher's LSD correction has been used. However the alpha error 0.025 reported is due to a Bonferroni's correction; indeed the Fisher's LSD correction used by the authors is not taking into account that multiple comparisons will be performed and it's good only for the calculation of the effect size (see Anthony J. Hayter. The maximum familywise error rate of fisher's least significant difference test. Journal of the American Statistical Association, 1986, 81(396):

1000-1004, doi: 10.1080/01621459.1986.10478364).

[Editors' note: further revisions were requested prior to acceptance, as described below.]

Thank you for resubmitting your work entitled "Registered report: Melanoma exosomes educate bone marrow progenitor cells toward a pro-metastatic phenotype through MET" for further consideration at *eLife*. Your revised article has been evaluated by Charles Sawyers (Senior Editor), Michael Green (Reviewing Editor), and four reviewers, one of whom, John Harris, has agreed to reveal his identity.

The reviewers have discussed the reviews with one another and the Reviewing Editor has drafted this decision to help you prepare a revised submission.

The reviewers felt that the revised manuscript had been considerably improved. However, two major problems remain. First, given the aim of this project, it is of paramount importance that all details of the experimental design be reproduced as described in Peinado et al., Nature Medicine, 2012. Therefore, all of the reviewers feel strongly that exosome purification needs to be done by sequential ultracentrifugation as in Peinado et al. (see point 1). Second, the conflict of issue has not been adequately resolved (see point 2).

Essential revisions:

1) We appreciate the authors' efforts to address my concerns. However, the new experiment (protocol 3) added to the manuscript proposing to compare ExoQuick extraction with the sequential ultracentrifugation presented in the original publication, is not appropriate for the "Reproducibility project". This is a new experiment that can be the focus of a new manuscript. Even for such a method comparison manuscript, to indeed validate their side-by-side comparison of ExoQuick and ultracentrifugation, the authors should perform unbiased analyses, specifically proteomic analysis of exosomes by mass spectrometry, not just western blot analyses. For the purpose of reproducing the data in the original Peinado et al. publication, the original exosome purification method should be used. Despite the fact that the differences from the original study are clearly stated for each protocol, for Protocols 2 and 4, the authors are still relying ONLY on ExoQuick for exosome purification. Given that during exosome isolation using ExoQuick, contaminants (e.g. VEGF) are trapped by ExoQuick, and the fact that education would not be strictly and solely due to exosomes, we believe that all the experiments proposed in this paper should be performed using the sequential ultracentrifugation, not ExoQuick. Moreover, the authors are using System Biosciences ExoFBS for all the experiments, rather than depleting exosomes from the FBS using sequential ultracentrifugation.

2) The authors declared their conflict of interest as employees of System Biosciences, the company that produces ExoQuick as well as ExoFBS. However, the determination of the authors to use System Biosciences reagents to conduct this study (even in the context of comparing methods) is impacted by the conflict of interest and driven by their desire to promote System Biosciences products. Therefore, because of their vested interest, employees of SBI should be excluded from the author list of this reproducibility project or the manuscript should not use any System Biosciences products.

3) The authors included the scrambled short hairpin control (shScramble) B16-F10 cells as s control for the B16-F10 shMET experiments. However, the authors plan to rely solely on puromycin resistance to select the knockdown clones, but this is not sufficient to isolate a pure population of transduced cells that maintains high levels of vector integration and expression. The authors should evaluate the knockdown after 28 days of puromycin selection. The authors should either use fluorescence-activated cell sorting to select a pure population or they should maintain the cells under puromycin selection. Just as another control, it would be desirable to compare both B16-F10 wild-type and Scramble in order to verify that the scramble does not have any undesirable effects, since infection can cause genome instability that way would be a perfect control.

4) The synthetic liposome control was a bit confusing. The authors say they will use 5 µg to match the protein on exosomes, but there should not be any protein on synthetic liposomes. We think they should figure out a vesicle concentration based on NanoSight and then use the same concentration of synthetic liposomes.

---

## [Author Response]

1) Given the aim of this project, it is of paramount importance that all details of the experimental design be reproduced as described in Peinado et al., Nature Medicine, 2012. In particular, there are some critical differences in the experimental procedures of the proposed study that may lead to big differences in the results compared to the original Peinado et al. study. First and foremost, the method proposed to isolate exosomes for western blot and functional studies, as well as for the depletion of exosomes from FBS is based on ExoQuick extraction rather than sequential ultracentrifugation as in the original publication. This is a major issue that can hinder the reproducibility of the data, as during exosome isolation using ExoQuick, contaminants (e.g. VEGF) are trapped by ExoQuick, therefore education would not be strictly and solely due to exosomes. Clearly the authors are aware of this issue, as they have highlighted this difference. The main problem is that 1 μg of exosomal protein from ExoQuick-isolated exosomes would be the equivalent of 0.1-0.5 μg of exosomes isolated by classical ultracentrifugation (UC), as in the Peinado et al. study. This 10-fold difference in the amount of exosome protein delivered in education studies would lead to "under-education". Since in the Peinado et al. study the exosomal preparations were free of contaminants, as demonstrated by electron microscopy and western blot, and the authors' likely carry significant contaminants, a side by side comparison of ExoQuick and ultracentrifugation would be required to determine the amount of ExoQuick exosomal preparation equivalent to 10 μg of exosomes. A reference is not sufficient, actual data needs to be presented for this comparison, including quantitation and electron microscopy studies. The authors would have to measure the yield of exosomes, proteins and contaminants from both methods in parallel.

There are numerous publications presenting data comparing exosomal isolation methods that support the use of ultracentrifugation versus polymer-based precipitating methods such as ExoQuick. In particular, three papers demonstrate that up to 23 times more material is precipitated with ExoQuick compared to standard serial UC (Van Deun J et al., Journal of Extracellular Vesicles, 2014, 18(3), PMID: 25317274; Zlotogorski-Hurvitz A et al., J Histochem Cytochem. 2015, 63(3): 181-9, PMID: 25473095; Yamada T et al., J Vet Med Sci. 2012, 74(11):1523-5, PMID: 22785357). A recent study provides electron microscopy pictures that demonstrate that the higher protein yield obtained with the ExoQuick precipitation method compared to the UC gold standard is due to contaminants such as large protein aggregates (Sáenz-Cuesta M et al., Front Immunol, 2015 Feb 13;6:50; PMID: 25762995).

We have included a comparison of the two methods, ultracentrifugation and ExoQuick, as a pilot experiment (protocol 3) to occur prior to the in vivo experiment. This will compare protein yield, exosome number, and contaminants. This will allow for an adjustment, if needed, in the amount of isolated protein used in the education study to control for any potential differences in the number of exosomes relative to protein content isolated by each method. This will allow for the proposed education experiment to use an equivalent number of exosomes from 5 µg of total protein when generated by ultracentrifugation.

2) Importantly, the fact that the first three authors on the proposed Reproducibility project study are employees of System Biosciences, the company that produces ExoQuick as well as ExoFBS raises a red flag and creates a huge conflict on interest in terms of the choice of methodology. The choice of experimental approaches and reagents should be driven by the desire to perform the tests in conditions as close as possible to the original study, but the authors are clearly choosing a critical reagent that introduces a large variation from the original study based on financial interests.

To address the concern regarding the experimental approach, we have included a comparison between methods to the revised manuscript as described in point 1 above. Additionally, any known differences are stated a priori. We have also included in the conflict of interest statement that System Biosciences produces the two reagents in question as well as the TransDux and the CD63 antibody.*3) It is not clear whether the authors plan to use fresh or frozen (-20°C) exosomal preparations for education. There have been studies that have shown that freezing exosomes results in lysis of more than 50% of the preparation (Oosthuyzen et al., J Physiol. 2013,591(Pt 23): 5833-5842) which will affect the amount delivered in a single dose (if the exosomes were quantified prior to freezing and that amount is used for calculation of material to inject for education). Freezing could also affect the functionality of exosomes. The original Peinado et al. study had performed the education studies with fresh exosomes.*

The exosomes will be isolated, protein content determined, and used fresh for each injection. We have revised some of the language to attempt to make this clearer.

4) Another major problem is that the authors are using non-transduced B16-F10 cells as control for the B16-F10 shMET. It is highly recommended that a scrambled short hairpin control (shScramble) be used, as it is widely known that infection and puromycin treatment of cells may alter the cell population and the appropriate scrambled control in the same vector is available from the same company that produces the shMET. Importantly, the authors plan to rely solely on puromycin resistance to select the knockdown clones, but this is not sufficient to isolate a pure population of transduced cells that maintains high levels of vector integration and expression. In addition to verifying knockdown of MET, the authors also need to show that Met knockdown did not affect the growth of the cells (every integration event is unique and you never know where the lentivirus integrated).

We agree with this suggestion and have included the shScramble in the revised manuscript. It was not clear if it was used in Peinado et al. for Figure 4E where the cells were described as F10, not shScramble, unlike for the experiments reported in Figure 5 that describe the cells as shScramble.

The transduction efficiency (as monitored by GFP expression using fluorescent microscopy) will be recorded for each stable knockdown line and recorded. While sorting of the cells by FACS will not occur, the knockdown efficiency by western blot (protocol 2) will be compared to the original study to compare the achieved knockdown of the original and replication attempts. The revised manuscript includes additional details commenting on how detailed notes regarding cell passaging will be made available to assess if there are differences in cellular growth.

5) In protocol 1: The authors propose to use puromycin selection to generate stable B16-F10 shMet transfectants instead of FACS sorting GFP-positive cells, which should be sufficient. Upon reviewing the original paper, we could not find any details how the authors generated B16-F10 shMet cells, and whether these were stable transfectants. Was this communicated directly by the authors of the study?

Yes, this information was communicated directly by the authors of the original study. We have included additional text to highlight this.

6) It looks like they never tested whether MET was downregulated in exosomes themselves in supplementary Figure 5a, just in the parent cells from which they purified the exosomes. This is planned for the B16-F10 control and shMET exosomes, which we think is good. We would also suggest including B16-F1 exosomes, as there was a comparison by Western blot for the B16-F10 to B16-F1 exosomes in Figure 4a, at least that would be something to compare across studies.

Thank you for the suggestion. We had originally thought of including the B16-F1 exosomes as well, but considering the B16-F1 cells are not utilized elsewhere in the manuscript, we have not included it. While it would further help show that Met, and pMet, is present in B16-F10 cells with lower levels in B16-F1 cells, the shMet condition serves as the proper control for this. Additionally, since the B16-F1 and B16-F10 exosome comparison was not quantified in the original study, it is difficult to ascertain the relative level of Met or pMet in either cell line.

7) In the original paper there is a microarray analysis where genes with a fold change greater than 2 are reported. Apparently p-values from a t-test were computed with a permutation approach but not reported. It would be interesting to repeat also this experiment reporting the statistical significant genes controlling the false discovery rate.

We agree the microarray experiment would be interesting to replicate, however these types of experiments are excluded from all articles. These exclusion criteria are outlined on the project page (https://osf.io/e81xl/wiki/studies) and in a Feature Article describing the project (http://elifesciences.org/content/3/e04333). We understand that the exclusion of certain experiments limits the scope of what can be analyzed by the project, but we are attempting to identify a balance of breadth of sampling for general inference with sensible investment of resources on replication projects to determine to what extent the included experiments are reproducible. As such, we will restrict our analysis to the experiments being replicated and will not include discussion of experiments not being replicated in this study.

8) In the statistical analyses plan of the report it is claimed that Fisher's LSD correction has been used. However the alpha error 0.025 reported is due to a Bonferroni's correction; indeed the Fisher's LSD correction used by the authors is not taking into account that multiple comparisons will be performed and it's good only for the calculation of the effect size (see Hayter, 1986).

The Bonferroni correction (making the alpha error 0.025) was to account for the two measurements (lung photon flux and bones photon flux) on the same animals since they will be performed using two different test families (ANOVA and Kruskal-Wallis). However, within each measurement and test family alpha error of 0.025, the comparisons use a Fisher’s LSD correction. We agree with the reviewers comment on the use of a correction, such as Bonferroni or the modification of LSD by Hayter, as ways to control for the MFWER, however as Hayter describes in his 1986 paper, this applies in situations where the ANOVA is unbalanced or with a balanced design with four or more populations. Since the proposed analysis is balanced with three population groups, the LSD is sufficiently conservative and powerful to account for the multiple comparisons in this specific situation. This is further explained by Levin et al., 1994 and discussed in Maxwell and Delaney, 2004 (Chapter 5) and Cohen, 2001 (Chapter 12).

References:

Levin, J.R., Serline, R.C., & Seaman M.A. (1994). A controlled, powerful multiple-comparison strategy for several situations. Psychological Bulletin, 115, 153-159.

Maxwell, S.E. & Delaney, H.D. (2004). Designing experiments and analyzing data: a model comparison perspecitive. Lawrence Erlbaum Associates, Mahwah, N.J., 2nd edition.

Cohen, B.H. (2001). Explaining psychological statistics. John Wiley and Sons, New York, 2nd edition.

[Editors' note: further revisions were requested prior to acceptance, as described below.]

Essential revisions:

*1) We appreciate the authors' efforts to address my concerns. However, the new experiment (protocol 3) added to the manuscript proposing to compare ExoQuick extraction with the sequential ultracentrifugation presented in the original publication, is not appropriate for the "Reproducibility project". This is a new experiment that can be the focus of a new manuscript. Even for such a method comparison manuscript, to indeed validate their side-by-side comparison of ExoQuick and ultracentrifugation, the authors should perform unbiased analyses, specifically proteomic analysis of exosomes by mass spectrometry, not just western blot analyses. For the purpose of reproducing the data in the original Peinado et al. publication, the original exosome purification method should be used. Despite the fact that the differences from the original study are clearly stated for each protocol, for Protocols 2 and 4, the authors are still relying ONLY on ExoQuick for exosome purification. Given that during exosome isolation using ExoQuick, contaminants (e.g. VEGF) are trapped by ExoQuick, and the fact that education would not be strictly and solely due to exosomes, we believe that all the experiments proposed in this paper should be performed using the sequential ultracentrifugation, not ExoQuick. Moreover, the authors are using System Biosciences ExoFBS for all the experiments, rather than depleting exosomes from the FBS using sequential ultracentrifugation.*

The revised manuscript includes the approach to purify exosomes by sequential ultracentrifugation instead of ExoQuick and as well as use FBS depleted of exosomes by ultracentrifugation as described in the original paper and confirmed by the original authors. Additionally, the protocol to compare ExoQuick to ultracentrifugation has been removed since it is no longer necessary for this replication attempt.

*2) The authors declared their conflict of interest as employees of System Biosciences, the company that produces ExoQuick as well as ExoFBS. However, the determination of the authors to use System Biosciences reagents to conduct this study (even in the context of comparing methods) is impacted by the conflict of interest and driven by their desire to promote System Biosciences products. Therefore, because of their vested interest, employees of SBI should be excluded from the author list of this reproducibility project or the manuscript should not use any System Biosciences products.*

We hope the reviewers agree that with the above revisions the conflict of interest has been adequately resolved. There are still two reagents used in the study that are produced by System Biosciences, TransDux, which replaces polybrene during the transfection procedure, and the rabbit anti-CD63 antibody, which was added as an additional quality control measure not included in the original study. Both are listed in the competing interests statement. As a reminder, TransDux reagent is similar to polybrene and enables high transduction rates of virus into most cells, sometimes at higher rates than polybrene. Most importantly, the knockdown efficiency, which is partly reflected by the infection reagent used, will be reported for both cells and exosomes, and for at the least the cells, will be compared to the originally reported knockdown efficiency.

*3) The authors included the scrambled short hairpin control (shScramble) B16-F10 cells as s control for the B16-F10 shMET experiments. However, the authors plan to rely solely on puromycin resistance to select the knockdown clones, but this is not sufficient to isolate a pure population of transduced cells that maintains high levels of vector integration and expression. The authors should evaluate the knockdown after 28 days of puromycin selection. The authors should either use fluorescence-activated cell sorting to select a pure population or they should maintain the cells under puromycin selection. Just as another control, it would be desirable to compare both B16-F10 wild-type and Scramble in order to verify that the scramble does not have any undesirable effects, since infection can cause genome instability that way would be a perfect control.*

The revised manuscript has the stable cells maintained in puromycin for at least 28 days prior to performing the knockdown evaluation. Additionally, the cells will remain under puromycin selection throughout maintenance of the cell lines, however when the cells are plated for experiments (for exosome purification) they will not include puromycin, which is similar to how exosomes were isolated in the original study.

We agree the addition of untransduced B16-F10 cells would be a valuable additional control to include. However, it is not feasible to include this extension to the original work since it is not possible to balance this additional aspect with the main aim of this project: to perform a direct replication of the original experiment(s).

*4) The synthetic liposome control was a bit confusing. The authors say they will use 5 µg to match the protein on exosomes, but there should not be any protein on synthetic liposomes. We think they should figure out a vesicle concentration based on NanoSight and then use the same concentration of synthetic liposomes.*

Thank you for highlighting this detail. In communication with the original authors we obtained the original amount of synthetic liposome controls injected and have added it to the revised manuscript. The amount of liposomes to inject is based on a theoretical ratio of 4:1 protein:L-α-phosphatidylcholine. With a formulation of liposomes that is 13 mg/ml L-α-phosphatidylcholine, 2.78 mg/ml cholesterol (7:3 molar ratio) the control injections will use 1.25 µg L-α-phosphatidylcholine to mimic the 5 µg of exosomes injected.